# Participants’ Bias in Disability Research on Family Quality of Life during the 0–6 Years Stage

**DOI:** 10.3390/bs13090753

**Published:** 2023-09-11

**Authors:** Carmen T. Francisco Mora, Alba Ibáñez-García, Anna Balcells-Balcells

**Affiliations:** 1Faculty of Psychology, Education and Sports Sciences, Ramon Llull University, 08022 Barcelona, Spain; carmenrm@blanquerna.url.edu (C.T.F.M.); annabb0@blanquerna.url.edu (A.B.-B.); 2Faculty of Education, Group of Research on Quality of Life, Health and Supports in Socioeducative Contexts (EDU-QoL), Universidad de Cantabria, 39005 Santander, Spain

**Keywords:** family quality of life, conceptualization, participants, research ethics, disability, family

## Abstract

Background: Over the past two decades, various research teams have designed and applied instruments to measure the quality of life of families with a member who has a disability. A recent systematic review on the state of the Family Quality of Life in early care identified that many of these studies collected data only from the mothers. The present study aimed to investigate whether there is a bias in participant selection in these types of studies. Method: A systematic review of the scientific literature was conducted in three databases—Scopus, Web of Science, Eric—from 2000 to 2022. A total of 72 empirical studies were identified. Results: The findings indicate that most studies examining the Family Quality of Life were based on the information of a single informant per family unit. The profiles of participants according to the research objective are quite similar. In one-third of studies, the authors reported that family members who participate cannot be represented by only mothers or one participant per household. Conclusions: Given the dynamic and collective nature of the construct, the application of a systemic approach is necessary.

## 1. Introduction

Bronfenbrenner [1] describes the importance of the family microsystem as a natural context for the development of children in permanent interaction with other environments or systems. In this theory, the relationships of people with environments and systems are represented on a daily basis, influencing our psychomotor, cognitive, moral, affective, and social development. 

The child is the main microsystem, along with family, friends, school, and neighbors, who are other microsystems. In the mesosystem are the interactions between two or more microsystems; for example, the relationship of the family with teachers or with neighbors. In the exosystem are, for example, the extended family and the media. They are environments in which boys and girls do not interact directly, but they do influence them. For example, the work of parents affects free time with their sons and daughters. Finally configured are the macrosystem, culture, customs, and values, among others.

From the family systems theory, each family is conceived as a complex social system in which the individuals that make up the family influence each other to form a family unit whose globality is greater than the sum of its parts [2]. 

According to Mandak et al. [3], in the context of the systemic and ecological theoretical framework and the principle of integrality, family research should be understood from a holistic perspective rather than focusing on individual subsystems. These authors also argued that families “have qualities and characteristics that cannot be understood by narrowly focusing on one or two family members” (p. 7). Other scholars have similarly emphasized a transformative paradigm that prioritizes listening to the voice of the family [4]. This approach is characteristic of the inclusive movement [5,6] and what Bruner called the “narrative turn” in social sciences [7,8,9,10], which highlights the importance of the representativeness of the participating subjects in the fields of epistemology and research ethics without solely depending on external sources of information. 

Scabini and Iafrate [11] found that family research presents a challenge associated with “the need to obtain information both at the level of the family group and in relation to the subsystems that compose it” (p. 230). Similarly, participatory family research studies [12,13] emphasize the active role of family members as research subjects. In this study, we adhered to the first and sixth principles of Citizen Science: “Citizen science projects actively involve citizens in scientific tasks that generate new knowledge or a better understanding” and “Citizen science represents a type of research like any other, with its limitations and biases that must be considered and controlled” [14] (p. 1).

In the field of disability, Family Quality of Life (FQoL) research is characterized by a plurality of approaches that focus on different aspects of research among families (narrative, participatory, and inclusive). The conceptualization of FQoL requires a specific theoretical framework [15,16], a distinct definition, and assessment methods, which are distinct from Individual Quality of Life (QoL). Specifically, Gardiner et Iarocci [17] and Fernandez et al. [18] proposed consideration of the principles of systems theories to investigate in the research of FQoL. Boelsma et al. [19], in a case study on longitudinal research and holistic approaches, have shown how FQoL changes in a family over the years. 

Francisco et al. [20], in a systematic review, identified that many of the selected studies collected data from only mothers and suggested the need for future research “to understand the ethical requirement that the methods used to address FQoL respect the holistic nature of the research” (p. 16). Although some studies have highlighted ethical implications in FQoL research [21,22], there is a need for further development in the area to ensure that the research methods take into account the family and its subsystems as a unit to avoid biases in information and address crucial epistemological and ethical implications.

Given the importance of considering all family members in research focused on the family as a unit of analysis, this study aims to identify the extent of participation of family members in QoL studies for families with children with disabilities aged between 0 and 6 years. To achieve this objective, the following research questions will be answered: 

Q1: What are the profiles of participants in FQoL research during the 0–6 years stage? 

Q2: Are there differences in the profiles of participants according to the research objective, such as (a) development of conceptualization, (b) development of assessment instruments for FQoL, or (c) applied research?

Q3: If any biases or limitations are identified, what perspectives do researchers provide in response? 

Focusing on the profiles of the participants in the most relevant research conducted on FQoL will allow us to know in greater depth whether the advances made to date are based on a family approach that brings together the perspective of all or almost all the members of the family or, on the contrary, are fundamentally nourished by the voice of one member of the family (usually the mother) who represents the whole unit. Knowing where we are in relation to this aspect, as well as the limitations identified by the researchers, will hopefully encourage the design of future studies to understand family opinion in a way that is more in line with a holistic approach.

## 2. Methods

This study draws on a systematic review conducted by Francisco et al. [20], which examined the scientific literature on QoL research for families with children who have disabilities aged 0 to 6 years. A total of 63 articles were selected and analyzed from the perspective of the construct of FQoL.

In this study, based on the results obtained in the systematic review up to August 2019, the bibliographic search in the Scopus, Web of Science, and Eric databases was extended from 2019 until August 2022 using the same keywords: “Family quality of life” or “Quality of family life”. We also decided to exclude the term ‘disability’ to maintain the same criteria and arguments as in the previous SR (i.e., to enrich our understanding of FQoL, to cover any type or diagnosis of disability, and to improve our current measurements on disability-related FQoL). The keywords have been searched using the criteria: title, abstract, and keyword. 

The inclusion criteria for this study were: (a)reported empirical work that included families of children aged 0 to 6 years with disabilities and/or developmental concerns;(b)published after 1999, the FQoL studies began in the year 2000 within the field of disability;(c)written in English, most commonly used in publications, or Spanish, which is the researchers’ native language;(d)published in peer-reviewed journals or book chapters.

In terms of exclusion criteria were studies that: (a)considered disability as a disease since the new paradigm of FQoL and early intervention focuses on the family rather than on the disease, according to the medical-rehabilitative paradigm;(b)examined FQoL from an individual rather than a holistic perspective;(c)conceptualized FQoL from a medical-rehabilitative perspective. The medical-rehabilitative perspective focuses attention on the child’s disability rather than on the unity of the family and respecting its protagonism. The exclusion criterion “considered disability as a disease” refers to the exclusion of those articles whose understanding of disability is based on the medical model, according to which disability is “the restriction or lack (due to an impairment) of the ability to of the ability to perform an activity in the manner or within the range considered normal for a human being” [23]. We argue with the World Health Organization, which in 2001 defined disability as “an umbrella term for impairments, activity limitations, and participation restrictions. It denotes the negative aspects of the interaction between an individual (with a health condition) and that individual’s contextual factors (environmental and personal factors)” [23].

The present study extends the systematic review by Francisco et al. [20], removing theoretical articles, and includes studies published between 2019 and 2022. A total of 1492 studies were identified, and after eliminating duplicates and applying the exclusion criteria (Figure 1), 72 articles were finally selected for the current systematic review.

A total of 72 articles were analyzed in this study, with 59 articles from the systematic review by Francisco et al. [20] and 13 from the updated search. The breakdown of these publications is as follows: (a) 71 empirical studies found in the databases; and (b) T = the Family Quality of Life Survey (FQOLS-2006) is identified through other sources [25].

## 3. Results

The 72 selected empirical studies were classified and analyzed following each research question.

### 3.1. What Is the Profile of Participants in FQoL Research during the 0–6 Years Stage?

The articles were evaluated based on those following a “traditional approach,” which, according to Wang et al. [26], are studies that only relied on the perspective of a single family member to assess FQoL and those that follow an approach that is consistent with the systemic theory. The FQoL research developed from that conducted in the individual QoL setting and in the belief that a qualified family member could provide information about the others. The systemic approach requires that the information come from family members due to their relational or subsystemic position in the family unit as a whole. 

As shown in Table 1, 65 out of the 72 selected articles followed the traditional approach, which indicates that each family unit surveyed or interviewed was represented by a single member or main caregiver. These studies are grouped in column (1) of Table 1 under the formula nF = nP, where n represents the number, F the families, and P the participants. The remaining seven studies are included under the formula nF < nP, suggesting that for each family, there were more participants from families surveyed or interviewed. Within this second group, distinctions included the study by Vanderkerken et al. [27], which presents a systemic approach by focusing on the family as a unit rather than on information provided by a single participant. In addition, four studies focused on the parental subsystem, with one study on the sibling subsystem and another study comparing the QoL of fathers and mothers.

Studies adopting a traditional approach were classified based on the profile of the single participant, typically specifying their gender and relationship to the child with a disability. To facilitate the analysis of the results, these studies were grouped into five categories: (1) mothers that constituted 50–69% of participants; (2) mothers that constituted 70–89% of participants; (3) mothers that constituted 90–99% of participants; (4) only mothers participated; and (5) no informant profile specified. In cases where only mothers were involved, or no information on gender or relationship was specified, the studies were sorted alphabetically. 

The percentages of the five groups of studies are stated below in descending order. The largest group (33% of the total studies) comprised 22 studies, with the participation of mothers ranging from 70% to 89%. The second group, comprising 24% of the total studies, was represented by 16 studies that did not specify the profile of the participant. The third group, consisting of 23% of the total studies, was represented by 15 studies with 90% to 99% participation of mothers. The fourth group comprised 11% of the total studies and corresponded to 11 studies, with 50% to 69% of mothers participating. Finally, 9% of the total studies were represented by six studies where only mothers participated. In five of the six studies, the researchers explicitly invited only mothers to participate, while in the remaining study [73], the authors invited a member representing their respective families to participate, but only 10 mothers agreed to participate.

The studies employing new approaches (n = 7) were organized into three groups: The first group comprised a single study by Vanderkerken et al. [27], which stands out as having utilized a systemic design representing all three subsystems: spousal, parental, and sibling. This study included 49 parental dyads, 1 father, 12 mothers, 10 children with disabilities, and 14 siblings, with a total of 135 participants from a sample of 63 family units (nF 63 < nP 135). The authors stated, “In addition, we went beyond parents’ perceptions by also taking into account the views of children (with and without disability) on FQoL and by comparing parents’ and children’s views on FQoL” [27] (p. 782).The second group consisted of five studies that investigated FQoL from the perspective of both fathers and mothers. These studies were grouped because the number of families was smaller than the number of participating relatives (nF < nP). Four of these studies adopted a parental subsystemic approach. Vanderkerken et al. [93] examined 34 parental dyads and argued that “members of the same family had different opinions regarding FQoL, which supports the idea of including every family member’s opinion when evaluating the complex reality of quality of life in families” [29] (p. 13). This systemic approach examines FQoL from the standpoint of the members of the parental subsystem. Similar approaches can be found in the studies by McStay et al. [91], Demchick et al. [94], and Mello et al. [92]. The fifth study in this group was by Wang et al. [26], which compared the individual perspectives of fathers and mothers. While providing information on the parental subsystem, the purpose of the study was to “test whether mothers and fathers similarly view the conceptual model” [26] (p. 977).The third group comprised a study by Moyson and Roeyers [95], which approached the assessment of QoL from the experiences of siblings, concluding that siblings may define their QoL differently from their parents. Based on the data analysis, only 5.47% of the reviewed studies adopted a systemic or subsystemic approach. These approaches were characterized by a detailed profiling of each participant’s position within the family system and an assessment of their respective perceptions in relation to those of other family members. Rather than assessing their knowledge, these studies focused on their roles as fathers or mothers, sons or daughters, brothers or sisters, grandfathers or grandmothers, or other family members. Participants’ gender identity and their dynamic interactions with other members of the family unit were considered key elements. Vanderkerken et al. [27] pointed out the primary strength of the systemic approach by stating, “Discussing and examining differences of opinion can be a valuable approach to generate a nuanced picture of life in a family” [27] (p. 750).

### 3.2. Are There Differences in the Profiles of Participants according to the Research Objective? 

To address the second research question, the 72 studies were categorized into four groups based on their research objectives (see Appendix A Table A1): (1) 7 studies aimed at developing instruments for assessing FQoL, including two international scales—the FQoL Scale [46]) and the FQoL Survey [25]—for all age groups, as well as five specific instruments designed to assess the QoL of families with children with disabilities aged 0–6 years; (2) 10 studies aimed at the validation of the instruments; (3) 32 studies either compared or described FQoL; and (4) 23 studies examined the predictors of FQoL. 

Figure 2 illustrates a graph linking the participant profile data (Table 1) with the four types of studies based on their research objectives.

The studies in groups 1 and 2 are characterized by the traditional approach and have minimally taken into account the perspectives of various family members. A significant difference was observed in the “unspecified” variable, where four studies that focused on the development of instruments for measuring FQoL did not identify the personal and family characteristics of the participants. For instance, the FQoL Survey was administered to the main caregiver of the person with a disability to report on the family’s QoL [25]. Other studies, such as those conducted by García-Grau [79], Giné [80], and Leadbitter et al. [82], omitted references to participant profiles for selection purposes.

Among the 32 studies in the third group, which had a comparative or descriptive nature of the FQoL, six studies stood out for utilizing a subsystemic approach with a focus on siblings of children with disabilities [95] or the parental dyad [22,91,92,93].

Finally, in the group of predictive studies on FQoL, only the study by Vanderkerken et al. [27] followed a systemic approach. Most studies in this group had a higher percentage of participating mothers (between 70% and 99%). 

### 3.3. If Any Biases or Limitations Are Identified, What Perspectives Do Researchers Provide in Response? 

To answer this question, the limitations identified by the authors in their respective studies were analyzed. Regarding coherence between the conceptualization of the construct and the methodological design, Boehm and Carter [53] explicitly stated that they initially relied on only one parent’s perceptions of FQoL. The definition of FQoL involved ‘‘a dynamic sense of well-being of the family collectively and subjectively defined and informed by its members, in which individual and family-level needs interact’’ [16] (p. 262). “Therefore, one parent’s perception of FQoL may not reflect the collective view of all members of the family” [53] (p. 111). 

However, the mentioned limitation was not observed in the 11 studies that incorporated the definition of Zuna et al. [16] into their theoretical frameworks [30,31,38,42,45,50,55,57,58,60,69,76]. This indicated a significant gap between the stated intentions of the researchers to study the family unit and the methods adopted to conduct the research [30,33].

While the authors of the selected studies did not expressly identify any reporting bias, the study by Wang et al. [26] revealed that fathers did not differ significantly from mothers in assessing their overall FQoL. However, this study relied on other authors [39,67,72] to justify their preference for the opinion of a single family member.

Nonetheless, 25 out of the 72 selected studies acknowledged a limitation in their research due to the involvement of only mothers or one participant per household, which can be implicitly understood as an information bias. There are several studies in which metaphors based on photography are used, where the image was expected to correspond to the photographed object, that is, be truthful. However, articles highlighting the limitations of the FQoL instruments that relied solely on mothers’ involvement without participation from the rest of the family reported that the truthfulness of the image would be compromised [42], the image would be unfocused [64], it would capture only half the picture [3], or in any case, it would be a ‘snapshot’ unable to reflect the dynamic nature of family life [27,33,60,86].

Another metaphor used by the authors was that of a choir to emphasize the necessity of acknowledging the diversity of familial voices, as a choir is composed of multiple voices. This concept was specifically highlighted in three studies that adopted a systemic or subsystemic approach [27,93,95]. Tait and Hussain [88] identified “the integrity, uniqueness, and complexity of individual experiences and perceptions of FQoL in their mixed methods study” [88] (p. 12).

To summarize the recognition of implicit reporting biases, six quotes were identified from studies that analyzed the family as the unit of analysis [26,30,33] and studies positing that the opinions and perceptions of other family members cannot be taken for granted [60,65,90,96]. The definition of FQoL entails a methodology that reflects collective information [27], and therefore, family members who did not participate cannot be represented by the informant. If the number of participants and family units were the same, the research would be compromised by information bias. The use of metaphors, such as the image or the chorus of voices, also highlights the need for a systemic approach that considers both the number of informants and their position or role in the family for examining a collective construct such as the family.

## 4. Discussion

In the following sections, the results are discussed in line with a holistic view that contextualizes the problems explored in relation to different theories, as well as the historical and cultural context in which research on FQoL has been conducted.

A relevant difference between the systemic approaches and traditional models is that traditional models often struggle to achieve a balance between assigning importance to a qualified profile, typically the mother while downplaying the role and gender of the sole participant by leaving it unspecified.

### 4.1. Scope of the Definition of FQoL Considering All Voices in the Family

The selection bias among participants becomes evident when examined in relation to the theoretical framework, the definition of FQoL, and the need to avoid assuming the perceptions and views of family members who were not involved in the research. A significant milestone in the conceptualization of FQoL was the definition of the construct by Zuna and colleagues [16], which identifies two essential characteristics of the FQoL construct: (1) the dynamicity of family relationships and (2) that it is collectively and subjectively defined. 

Regarding the first characteristic, none of the selected articles explicitly addressed the relationship between the dynamicity of the FQoL construct and the bias resulting from a single informant’s participation, which compromises the conceptualization of FQoL. However, this relationship may be implicitly conveyed through the image metaphor used by the aforementioned authors to refer to the FQoL construct.

Regarding the second characteristic of the definition of FQoL, only Boehm and Carter [53] examined the relationship between the limitation of the participation of one family member and the need for the FQoL construct to be defined subjectively and collectively.

However, none of the scales or questionnaires developed to measure FQoL have been designed following the epistemological requirement of representing the diverse subsystems that integrate the family unit. It is generally considered sufficient to obtain information from those who know the child best, whether with or without a disability (i.e., their primary caregivers, usually their mothers). However, this issue is not unique to FQoL research and is present in other scientific fields. Shah et al. [97] found that individual bias in the design and use of instruments to measure the QoL of both the family unit and its individual members is a widespread limitation, even when referring to the inclusion of the family as a whole.

To obtain significant information from a systemic standpoint, it is necessary to encourage the participation of representatives from each subsystem of the family, which is not evident in existing instruments, especially those where the role of informants is not specified. The only exception is the 2019 revision of the CdVF-ER user manual, which invited various family members to participate in the research. It was noted that “the scale should be answered by fathers, mothers, siblings, or legal guardians of persons with intellectual or developmental disabilities (IDD). In any case, it should always reflect family opinion” [97] (p. 6). Therefore, the CdVF-ER for children below 18 years of age was designed using a systemic approach by considering the role of children within the family unit.

While studies examining the quality of family life attempt to conceptualize this systemic construct, the use of “traditional” instruments that rely on information from a single participant can limit the representation of the entire family unit or other family members.

Alderesey et al. [98] described how interviews were conducted at doorsteps, which facilitated the participation of other family members and neighbors who joined in the conversation. The authors explained that these contributions “enriched the data I collected from my primary respondents and provided me with a greater understanding of family and household dynamics” [98] (p. 4) and argued that “participants in this study identified FQoL as a dynamic and subjective concept” [98] (p. 6). Qualitative studies, such as the one conducted by Boelsma et al. [19], show how the participation of the whole family in a longitudinal study makes it possible to identify different types of family dynamics that are very difficult to appreciate through the traditional functional approach. Hu et al. [99] pointed out that this traditional approach contradicts “the original target of measuring FQOL of the whole family unit. Consequently, researchers should take sufficient caution when analysing and generalising results to the family unit when only one or two family members have completed the instrument” [99] (p. 8).

The selected studies demonstrate an increasing tendency to consider the opinions of various family members beyond just the parents. Rillotta et al. [56] and Schlebusch et al. [34] suggested that future research should consider assessing FQoL from multiple perspectives, such as those of parents, siblings, grandparents, and individuals with disabilities. Vanderkerken et al. [27] also highlighted that “future research can explore strategies to also include the views of children under 12 (with or without disability) on FQoL, for example, by simplifying the instruments, adding pictures, etc.” [27] (p. 801). Their study is pioneering in this respect. Chinn and Balotta [100] encouraged the use of “creative” methodologies, such as photovoice, because they allow the artistic expression of the participants while involving the critical analysis of the researchers.

### 4.2. Epistemological Scope of New Approaches in Research 

An epistemological perspective allows for a better understanding of the systemic dynamics of the family [27], emphasizing the transformation of reality wherein disadvantaged people are placed at the center. Rodríguez-Cley et al. [101] aimed to “demonstrate how other epistemologies, coming from the experience of disability, can nurture participatory methodologies and design research” [101] (p. 26). This is in line with the ongoing “narrative turn” observed in various areas of research, which aims to empower people to exercise greater control over aspects of their own lives [102]. 

Zuna and colleagues [16] indicated that FQoL is not only collectively but also “subjectively” defined. While this plurality of “nuances” is positive, the subjective nature of its definition can pose challenges for researchers. In the field of QoL research, Brown and Schippers [103] cautioned against the traditional approach in that “the results of qualitative research often carry less weight because of the strong influence of a traditionally positivist approach that considers qualitative data as subjective” [103] (p. 8). In addition, the authors highlighted the importance of “the scientifically recorded experiences and views of people with life problems and their families” [103] (p. 8) and insisted that data itself is never subjective, but rather “it is the interpretation that risks being subjective” [103] (p. 7).

Furthermore, the development of the ten principles of citizen science may contribute to a shift toward involving people as “contributors, collaborators, or as leaders in research projects” [14]. 

For example, Del Toro and Sanchez [104] counted “with the collaboration of 110 families from 20 early intervention centers in different autonomous communities in Spain” with the objective of “determining if there are significant differences between two groups formed by families that are working within the family-centered model and, on the other hand, other families that are not within this model” [104] (p. 15). The authors mentioned that they did not expect the result: “it is the families that do not work within the Family-Centered Model that present a higher quality of life in all the subscales analyzed with respect to the families that do work within this model” [104] (p. 18). Del Toro and Sanchez [104] concluded that: 

“Culturally, families have been mere observers of the work done with their children with functional diversity. It is essential to break this idea of passive work so that the family becomes an active part of the process. Only from this perspective will the family feel able to develop the model.”[104] (p. 19)

A cultural approach is also seen in studies comparing the QOL of Latino and non-Latino families in the United States [72,73] as well as in the adaptations of the scales in a multitude of countries. 

Another significant aspect of citizen science is the democratization of scientific research and knowledge. In the context of FQoL research, it is important for family members to establish a deep and existential connection with the research topic, which may encourage them to participate responsibly. 

### 4.3. Ethical Scope of New Approaches in Research 

The limitations observed in the conceptualization of FQoL not only manifest in the epistemological and methodological aspects but also indicate the existence of ethical considerations. First, the ethical dimension emphasizes the importance of inclusive research that amplifies the voices of the most vulnerable people within the family unit, such as minors and people with disabilities [105,106].

Second, the ethical need to represent vulnerable individuals may lead researchers to explore suitable methods to enable them to be active participants in the research. For instance, the photovoice methodology has been examined as a means of providing visibility to vulnerable members of society while respecting their privacy [107,108,109]. In photovoice research, participants are informed of the ethical aspects of the use of images [110], and the informed consent of the individuals appearing in the photographs is requested. In FQoL research, it may also be ethically advisable to obtain informed consent or assent from family—so that participants can speak on their behalf.

In addition, recognizing individuals as active subjects in their own lives and striving to capture the authentic voices of families may encourage researchers to involve as many family members as possible in the research process, ensuring that the different family subsystems are adequately represented. 

Regarding future research directions, researchers in the field of FQoL should embrace its conceptualization as a family unit with unique methodological and temporal needs, as opposed to understanding QoL solely from the perspective of individual family members. Egaña and Barría [111] highlighted the need to “be rigorous with the idea that the family as a unit of analysis is different from the individuals who, as family members, compose it” (p. 15). Each researcher should, therefore, establish a theoretical framework relevant to the specific objects of their study, whether it be the entire family, individual family members, or subsystems, to ensure consistency in the methodological design. Studies using mixed methodologies are also noteworthy [49,53,73,85]. Longitudinal studies have also been recommended for FQoL research [41,89]. In particular, Gardiner and Iarocci [17] pointed out the need for longitudinal research when examining the dynamic interactions between individual and family needs. This approach may also apply to research on family routines, considering their dynamic nature in the context of FQoL [31].

In addition, FQoL research may benefit from the epistemological and ethical approaches that are being applied in the field of social sciences, such as the inclusion of persons with disabilities, citizen participation, and respect for the privacy of those being studied.

The most significant limitation of our analysis and discussion has been based on the content of the articles, knowing that researchers often do not explain in detail important aspects of the methodological process.

## 5. Conclusions

Based on the systematic review of FQoL research, it can be concluded that, with the exception of the study by Boehm and Carter [53], the remaining 71 articles reviewed did not consider the epistemological need inherent in the FQoL construct for the participation of all family members. This inference is drawn from the observation that the epistemological limitation was not taken into account either because of the lack of a theoretical framework, the definition of FQoL, or inconsistency with the requirements of the integrated definition in their respective theoretical frameworks. In addition, the relationship between this limitation and the dynamic nature of the construct has not been examined. 

The findings of the current research suggest that only a few of the reviewed studies have adopted a systemic or subsystemic approach (8.2%) compared to the traditional approach, where only one member per family unit participates, which was employed by the majority (91.8%) of the reviewed studies. Approximately one-third of the selected studies identified the potential for possible reporting biases due to the participation of only one family member.

On evaluating the “limitations” section of some of the selected studies, it was noted that potential participants’ bias may affect the validity of the results by assuming that the opinions and knowledge of the different members of the family are accurately represented by one family member. Recognizing and addressing these considerations is crucial for upholding the ethics of scientific research.

In summary, this research holds significant implications in the field of FQoL in social, health, and education research. The findings highlight the importance of ensuring the participation of the family as a collective unit and recognizing the unique perspectives and contributions of each individual member. 

## Figures and Tables

**Figure 1 behavsci-13-00753-f001:**
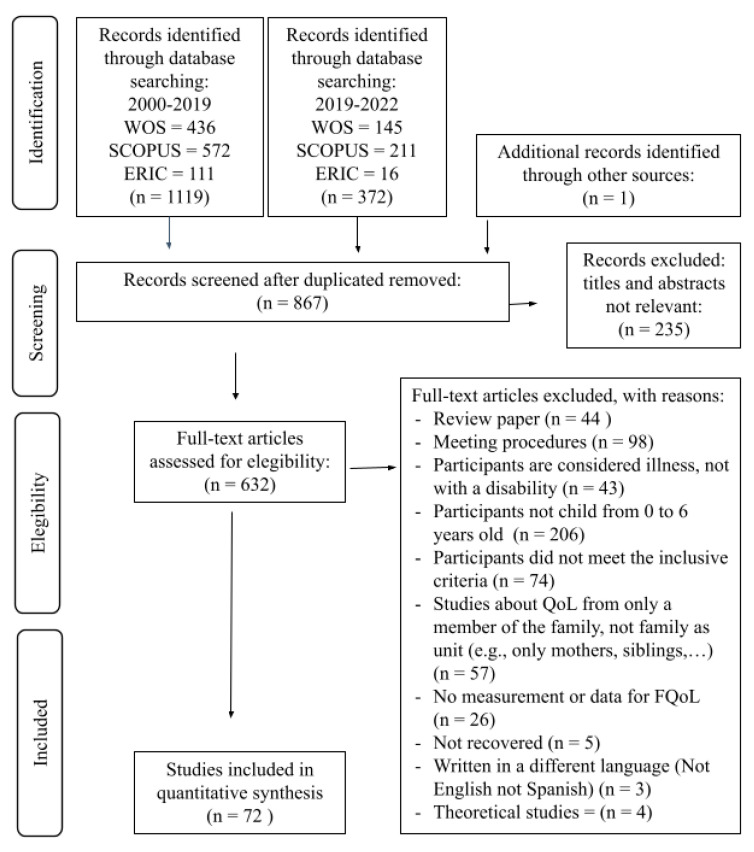
Flowchart of the selection process according to the recommendations of the Prisma statement [24]. Source: Own elaboration.

**Figure 2 behavsci-13-00753-f002:**
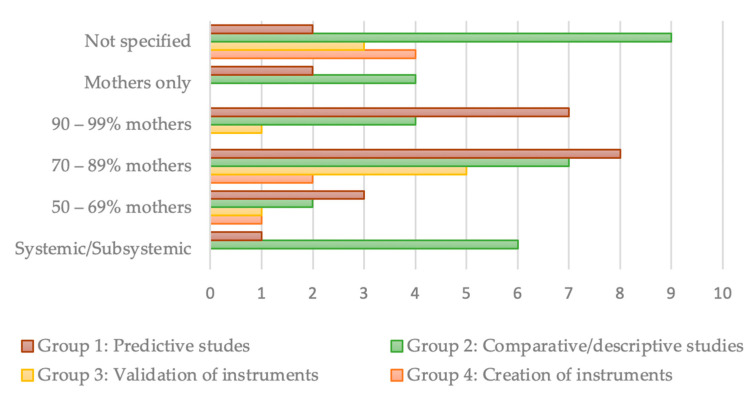
Participant profile data by types of studies.

**Table 1 behavsci-13-00753-t001:** Articles classified according to participant profile.

Approach	Participants	Number of Studies	Authors
Traditional Approach (nF = nP)	50–69% mothers	7	Rivard et al. [28]; Feng et al. [29]; Huang et al. [30]; Mas et al. [31]; Levinger et al. [32]; Schlebusch et al. [33]; Schlebusch et al. [34].
	70–89% mothers	22	Svavarsdottir and Tryggvadottir [35]; Córdoba et al. [36]; Verdugo et al. [37]; Chiu et al., [38]; Waschl et al. [39]; Escorcia et al. [40]; Giné et al. [41]; Balcells-Balcells et al. [42]; Balcells-Balcells et al. [43]; Bhopti et al. [44]; Chiu et al. [45]; Hoffman et al. [46]; Algood and Davis [47]; Barnard et al. [48]; Balcells-Balcells et al. [49]; Eskow et al. [50]; Hsiao et al. [51]; Hsiao et al. [52]; Boehm y Carter [53]; Samuel et al. [54]; Schertz et al. [55]; Rillotta et al. [56].
	90–99% mothers	12	Kyzar et al. [57]; Kyzar et al. [58]; Jackson et al. [59]; Taub y Werner [60]; Samuel et al. [61]; Susanto et al. [62]; Samuel et al. [63]; Steel et al. [64]; Epley et al. [65]; Davis y Gavidia Payne [66]; Summers et al. [67]; Clark et al. [68]
	Mothers only	6	Cohen et al. [69]; Holloway et al. [70]; McStay et al. [71]; Meral et al. [72]; Rodrigues et al. [73]; Valverde y Jurdi [74]
	Not specified	18	Bello-Escamilla et al. [75]; Brown et al. [22]; Brown et al. [76]; García Grau et al. [77]; García Grau et al. [78]; García Grau et al. [79]; Giné et al. [80]; Hielkema et al. [81]; Leadbitter et al. [82]; Lei et al. [83]; Liu et al. [84]; Lee et al. [85]; Neikrug et al. [86]; Perry e Isaacs [87]; Tait and Husain [88]; Tejada-Ortigosa et al. [89]; Verger et al., [3]; Wang et al. [90].
New approaches (nF < nP)	Systemic	1	Vanderkerken et al. [27].
	Dyads fathers/mothers	5	Wang et al., [26]; McStay et al., [91]; Mello et al. [92]; Vanderkerken et al. [93]; Demchick et al. [94]
	Siblings	1	Moyson et Royers [95].

Source: Own elaboration.

## Data Availability

Data sharing is not applicable.

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
