# Peer review of "Participants’ Bias in Disability Research on Family Quality of Life during the 0–6 Years Stage"

_behavsci, 2023, doi:10.3390/bs13090753_

Round 1

Reviewer 1 Report

General Comments

Interesting topic and well worth the research, but some issues need to be addressed (see below).

Specific Comments

Abstract – Please list the 3 databases that was included in your literature search.

Introduction – The intro does not include any reference to the problem mentioned in the abstract: “recent systematic review on the state of art of Family Quality of Life in early 14 care identified that many of these studies collected data only from the mothers”. This seems to be an important justification for your research, please include it.

Introduction – The intro also does not reference anything about disability, although it is mentioned in the abstract and title. Please include something about disability.

Methods – Was a framework or standard used to plan the review? E.g., PICO, etc.? If so, please report on how you addressed each part of this framework.

Methods – How did you define disability for this review?

Methods – The exclusion criterion “considered disability as a disease” is important to discuss. Please elaborate further to justify why this was an exclusion criterion for clarification to future readers.

Discussion – While this section is interesting and bring up valuable points, it would be improved with the addition of subheadings to allow for easier reading. As it stands, it goes from one point to another without a clear logical flow.

Discussion – There is no review limitations section, please add this.

Author Response

Response to Reviewer 1 Comments

GENERAL COMMENTS:

Point 1: Interesting topic and well worth the research, but some issues need to be addressed (see below).

Response 1: We are sincerely grateful for the time spent in re-examining this study. We believe that our manuscript has greatly improved thanks to the reviewers’ suggestions and is now more appropriate to be published by the journal Behavioral Sciences.

Bellow are our answers to each of your interesting suggestions.

SPECIFIC COMMENTS

Point 2: Abstract – Please list the 3 databases that was included in your literature search.

Response 2: In the abstract, we have listed the 3 databases that was included in our literature search.

The text has been reformulated as follows: “A systematic review of the scientific literature was conducted in the 3 databases -Scopus, Web of Science, Eric- from the year 2000 to 2022”. 

Point 3: Introduction – The intro does not include any reference to the problem mentioned in the abstract: “recent systematic review on the state of art of Family Quality of Life in early 14 care identified that many of these studies collected data only from the mothers”. This seems to be an important justification for your research, please include it.

Response 3: We have included the text: “Francisco et al. [20] in a systematic review identified that many of the selected studies collected data from mothers only and suggested the need for future research "to understand the ethical requirement that the methods used to address FQoL respect the holistic nature of the research" (p. 16)” in the Introduction.

Point 4: Introduction – The intro also does not reference anything about disability, although it is mentioned in the abstract and title. Please include something about disability.

Response 4: In the introduction, we have mentioned that our research on FQoL was conducted in the field of disability.

Point 5: Methods – Was a framework or standard used to plan the review? E.g., PICO, etc.? If so, please report on how you addressed each part of this framework.

Response 5: We have clarified the method in the direction you have suggested.

Point 6: Methods – How did you define disability for this review?

Response 6: Our definition of disability in this review is that of the WHO (2001). The ICF defines disability as “…an umbrella term for impairments, activity limitations and participation restrictions. It denotes the negative aspects of the interaction between an individual (with a health condition) and that individual’s contextual factors (environmental and personal factors). The definition has been introduced in the text (method-exclusion criterion).

Point 7: Methods – The exclusion criterion “considered disability as a disease” is important to discuss. Please elaborate further to justify why this was an exclusion criterion for clarification to future readers.

Response 7: The exclusion criterion “considered disability as a disease” refers to the exclusion of those articles whose understanding of disability is based on the medical model, according to which disability is “the restriction or lack (due to an impairment) of the ability to of the ability to perform an activity in the manner or within the range considered normal for a human being” (OMS, 1980). We have clarified this exclusion criteria in the text.

Point 8: Discussion – While this section is interesting and bring up valuable points, it would be improved with the addition of subheadings to allow for easier reading. As it stands, it goes from one point to another without a clear logical flow.

Response 8: We have clarified de Discussion with the addiction of the following subheading in the text:

a) Scope of the definition of FQoL considering all voices in the family

b )Epistemological scope of new approaches in research;

c) Ethical scope of new approaches in research.

Point 9: Discussion – There is no review limitations section, please add this.

Response 9: We have added a review limitations paragraph.

Reviewer 2 Report

TITLE AND ABSTRACT

The title of the article is informative and clearly states the topic of the study. The abstract provides a brief overview of the study's background, methods, and key findings.

INTRODUCTION

·         Contextual Background: The introduction provides a concise background on the importance of family research from a holistic perspective, emphasizing the need to consider the family as a unit rather than focusing solely on individual members. However, it could be more detailed and include specific examples or studies that have adopted this holistic approach to highlight its significance in the field.

·         Theoretical Framework: The introduction mentions the importance of a theoretical framework for FQoL research but does not elaborate on the specific theories or frameworks commonly used in the field. Consider briefly introducing the systemic and ecological theoretical framework and its relevance to the study.

·         Missing Objectives: The introduction mentions three research questions (Q1, Q2, Q3), but it does not explain why these questions are essential or what the study aims to achieve through answering them. Provide a clearer rationale for the research questions and their significance in advancing the understanding of FQoL research.

METHODS AND MATERIALS

·         Lack of Clarity in Search Period: The method mentions extending the bibliographic search until August 2022 using the same keywords as a previous systematic review conducted by Francisco et al. [18]. However, it is unclear whether the search period was limited to the time between the initial systematic review in 2019 and August 2022 or if it also included the period covered by the initial review. Clarify the exact time frame for the extended search to provide transparency and avoid confusion.

·         Inclusion and Exclusion Criteria: The method outlines the inclusion and exclusion criteria for selecting studies but does not explain the rationale behind each criterion. Provide a brief explanation for each criterion to justify its relevance in the context of the research objective.

·         Justification for Exclusion Criteria: The exclusion criteria seem appropriate, but the method could elaborate on why certain criteria, such as considering disability as a disease or examining FQoL from a medical-rehabilitative perspective, were chosen. Providing clear reasons for these exclusions will enhance the rigor of the study.

RESULTS

·         Limited Analysis of New Approaches: The section briefly discusses studies adopting new approaches (systemic or subsystemic) but lacks in-depth analysis and comparison with the traditional approach. A more comprehensive comparison of the findings from the different approaches and their implications for the research is necessary to highlight the importance of adopting a systemic perspective in family research.

DISCUSSION

The discussion presented in this section offers valuable insights into the complex and multifaceted nature of FQoL research. The authors provide a comprehensive analysis of the differences between systemic and traditional approaches in studying FQoL, shedding light on the limitations and potential biases in the current research methodologies. They also highlight the importance of embracing a holistic view that considers historical, cultural, and theoretical contexts in FQoL studies.

One of the strengths of this section is the critical examination of the limitations of traditional approaches that rely on single informants, typically mothers, to assess FQoL. The authors correctly point out that this approach may not fully capture the collective and subjective nature of FQoL as defined by Zuna et al. [15]. The discussion of the need for a systemic perspective and the inclusion of multiple family members' views to achieve a more comprehensive understanding of FQoL is well-founded.

Additionally, the authors acknowledge the ethical considerations in FQoL research, particularly the importance of representing vulnerable family members, such as minors and individuals with disabilities. The emphasis on obtaining informed consent or assent from family members and involving them actively in the research process aligns with ethical standards.

To enhance this section further, a few suggestions are proposed:

·         Provide Concrete Examples: To illustrate the differences between systemic and traditional approaches, it would be beneficial to include specific examples from previous FQoL studies. By analyzing and discussing real case studies, readers can better grasp the practical implications of these approaches.

·         Address Limitations of the Study: While the authors critique the limitations in current FQoL research, they should also acknowledge any potential limitations in their own study. Providing transparency about the limitations of the research and their potential impact on the findings would enhance the credibility of the paper.

·         Propose Practical Solutions: While the authors mention the need for systemic approaches and the inclusion of various family members, they could further explore practical ways to implement these suggestions in future research. Offering specific methodological strategies or potential study designs that incorporate a systemic perspective would be beneficial.

·         Discuss the Impact of Cultural Differences: The authors acknowledge the historical and cultural context in FQoL research, but they could delve deeper into how cultural differences may influence FQoL perceptions and outcomes. Recognizing and exploring these cultural nuances would enrich the discussion.

·         Highlight Research Gaps: Identifying areas where further research is needed could provide a launching point for future studies in the field of FQoL. By addressing research gaps, the authors can contribute to the advancement of knowledge in this domain.

CONCLUSIONS

The conclusions drawn from the systematic review of FQoL research offer valuable insights into the current state of the field. The identification of the epistemological limitation in the majority of reviewed studies is a critical finding, indicating a lack of consideration for the participation of all family members in the research process. The low adoption rate of systemic or subsystemic approaches in the reviewed studies also highlights a significant gap in the application of more inclusive methodologies.

Author Response

Response to Reviewer 2 Comments

We are sincerely grateful for the time spent in re-examining this study. We believe that our manuscript has greatly improved thanks to the reviewers’ suggestions and is now more appropriate to be published by the journal Behavioral Sciences.

These are our answers to each of your interesting suggestions:

TITLE AND ABSTRACT

Point 1: The title of the article is informative and clearly states the topic of the study. The abstract provides a brief overview of the study's background, methods, and key findings.

Response 1: Thank you for your appreciation.

INTRODUCTION

Point 2: Contextual Background: The introduction provides a concise background on the importance of family research from a holistic perspective, emphasizing the need to consider the family as a unit rather than focusing solely on individual members. However, it could be more detailed and include specific examples or studies that have adopted this holistic approach to highlight its significance in the field.

Response 2: Thank you for your appreciation. We have further detailed the background and included specific studies (p.e. Boelsma et al.[18]) that have adopted this holistic approach to highlight its significance in the field.

Point 3: Theoretical Framework: The introduction mentions the importance of a theoretical framework for FQoL research but does not elaborate on the specific theories or frameworks commonly used in the field. Consider briefly introducing the systemic and ecological theoretical framework and its relevance to the study.

Response 3: Thank you for your appreciation. We have briefly introduced the systemic and ecological theoretical framework and its relevance to the study.

Point 4: Missing Objectives: The introduction mentions three research questions (Q1, Q2, Q3), but it does not explain why these questions are essential or what the study aims to achieve through answering them. Provide a clearer rationale for the research questions and their significance in advancing the understanding of FQoL research.

Response 4: After the three research questions, we have included a paragraph to provide a clearer rationale for the research questions and their significance in advancing the understanding of FQoL research:

“Focusing on the profile of the participants in the most relevant research conducted on FQOL will allow us to know in greater depth whether the advances made to date are based on a family approach that brings together the perspective of all or almost all the members of the family, or on the contrary, are fundamentally nourished by the voice of one member of the family (usually the mother) who represents the whole unit. Knowing where we are in relation to this aspect, as well as the limitations identified by the researchers themselves, will hopefully encourage the design of future studies to understand family opinion in a way that is more in line with a holistic approach.”

METHODS AND MATERIALS

Point 5: Lack of Clarity in Search Period: The method mentions extending the bibliographic search until August 2022 using the same keywords as a previous systematic review conducted by Francisco et al. [18]. However, it is unclear whether the search period was limited to the time between the initial systematic review in 2019 and August 2022 or if it also included the period covered by the initial review. Clarify the exact time frame for the extended search to provide transparency and avoid confusion.

Response 5: We have clarified the search period (from 2000 to 2022).

Point 6: Inclusion and Exclusion Criteria: The method outlines the inclusion and exclusion criteria for selecting studies but does not explain the rationale behind each criterion. Provide a brief explanation for each criterion to justify its relevance in the context of the research objective.

Response 6: We have provied a brief explanation for each criterion to justify its relevance in the context of the research objective.

Point 7: Justification for Exclusion Criteria: The exclusion criteria seem appropriate, but the method could elaborate on why certain criteria, such as considering disability as a disease or examining FQoL from a medical-rehabilitative perspective, were chosen. Providing clear reasons for these exclusions will enhance the rigor of the study.

Response 7: We have provied a clear reasons for each exclusion criterion to enhance the rigor of the study.

RESULTS

Point 8: Limited Analysis of New Approaches: The section briefly discusses studies adopting new approaches (systemic or subsystemic) but lacks in-depth analysis and comparison with the traditional approach. A more comprehensive comparison of the findings from the different approaches and their implications for the research is necessary to highlight the importance of adopting a systemic perspective in family research.

Response 8: We have added a more comprehensive comparison of the findings from the different approaches and their implications for the research.

DISCUSSION

Point 9: The discussion presented in this section offers valuable insights into the complex and multifaceted nature of FQoL research. The authors provide a comprehensive analysis of the differences between systemic and traditional approaches in studying FQoL, shedding light on the limitations and potential biases in the current research methodologies. They also highlight the importance of embracing a holistic view that considers historical, cultural, and theoretical contexts in FQoL studies.

One of the strengths of this section is the critical examination of the limitations of traditional approaches that rely on single informants, typically mothers, to assess FQoL. The authors correctly point out that this approach may not fully capture the collective and subjective nature of FQoL as defined by Zuna et al. [15]. The discussion of the need for a systemic perspective and the inclusion of multiple family members' views to achieve a more comprehensive understanding of FQoL is well-founded.

Additionally, the authors acknowledge the ethical considerations in FQoL research, particularly the importance of representing vulnerable family members, such as minors and individuals with disabilities. The emphasis on obtaining informed consent or assent from family members and involving them actively in the research process aligns with ethical standards.

To enhance this section further, a few suggestions are proposed:

  • Provide Concrete Examples: To illustrate the differences between systemic and traditional approaches, it would be beneficial to include specific examples from previous FQoL studies. By analyzing and discussing real case studies, readers can better grasp the practical implications of these approaches.
  • Address Limitations of the Study: While the authors critique the limitations in current FQoL research, they should also acknowledge any potential limitations in their own study. Providing transparency about the limitations of the research and their potential impact on the findings would enhance the credibility of the paper.
  • Propose Practical Solutions: While the authors mention the need for systemic approaches and the inclusion of various family members, they could further explore practical ways to implement these suggestions in future research. Offering specific methodological strategies or potential study designs that incorporate a systemic perspective would be beneficial.
  • Discuss the Impact of Cultural Differences: The authors acknowledge the historical and cultural context in FQoL research, but they could delve deeper into how cultural differences may influence FQoL perceptions and outcomes. Recognizing and exploring these cultural nuances would enrich the discussion.
  • Highlight Research Gaps: Identifying areas where further research is needed could provide a launching point for future studies in the field of FQoL. By addressing research gaps, the authors can contribute to the advancement of knowledge in this domain.

Response 9: Thank you for your appreciation. Regarding your suggestions:

  • We have provided concrete examples to illustrate the differences between systemic and traditional approaches..
  • We have added limitations of our study.
  • We have proposed practical solutions.
  • We have briefly added the importance of consider cultural differences in FQoL research.
  • We have highlithed research gaps.
